# Juvenile and Osteoarthritic Human Chondrocytes Under Cyclic Tensile Strain: Transcriptional, Metabolic and Kinase Responses

**DOI:** 10.3390/ijms262210934

**Published:** 2025-11-12

**Authors:** Birgit Lohberger, Vincent Grote, Heike Kaltenegger, Dietmar Glänzer, Patrick Sadoghi, Tanja Kraus, Bibiane Steinecker-Frohnwieser

**Affiliations:** 1Department of Orthopedics and Traumatology, Medical University Graz, 8010 Graz, Austria; heike.kaltenegger@medunigraz.at (H.K.); dietmar.glaenzer@medunigraz.at (D.G.); patrick.sadoghi@medunigraz.at (P.S.); tanja.kraus@medunigraz.at (T.K.); 2Ludwig Boltzmann Institute for Arthritis and Rehabilitation, 5760 Saalfelden, Austria; 3Ludwig Boltzmann Institute for Rehabilitation Research, 1100 Vienna, Austria; vincent.grote@lbg.ac.at

**Keywords:** juvenile chondrocytes, osteoarthritis, mechanical stimulation, regulatory differences, metabolic network

## Abstract

Osteoarthritis (OA) involves cartilage breakdown and inflammation. This study compares juvenile and OA chondrocytes in gene expression, metabolism, and kinase activity, and tests mechanical stimulation to better understand cartilage health and degeneration. Juvenile (jCH) and OA (pCH-OA) primary chondrocytes were mechanically stimulated using the Flexcell™ FX5K system. Gene expression, protein phosphorylation, and metabolism were analyzed pre- and post-stimulation. Principal component analysis and effect size analyses identified molecular and signaling differences. Gene expression revealed significant differences between jCH and pCH-OA, with COL1 and RUNX2 upregulated in jCH, and MMP3 and ACAN downregulated. PCA revealed distinct expression patterns and marker correlations. Cyclic tensile strain affected biomarkers such as RUNX2, IL8, TLR4, BMP2, and MMP1 in a cell type-specific manner. Metabolic profiling indicated lower ROS and NAD^+^/NADH, and higher glutamate, lactate, and formate, with changes primarily driven by mechanical stimulation rather than cell type. Protein analysis showed altered AKT, STAT3, and MAPK phosphorylation, reflecting different mechanotransduction in healthy versus OA chondrocytes. Juvenile and OA chondrocytes show distinct molecular, metabolic, and signaling profiles, with mechanical stimulation driving key biomarker and metabolic changes. These differences highlight altered mechanotransduction in OA, providing insights into cartilage degeneration and potential therapeutic targets.

## 1. Introduction

Mechanical loading during joint movement is crucial for articular cartilage health, as it maintains a balance between matrix anabolism and catabolism [1]. Disturbance of this balance, particularly under excessive stress, contributes directly to the development and progression of osteoarthritis (OA) [2,3,4]. In this context, mechanical signaling plays a key role in regulating physiological and pathological processes in joint cells and tissues [5]. Over the years, research has shown that OA, as the most common degenerative joint disease, involves more complex processes than just “wear and tear” of the tissue as a response to mechanical stress. Mechanical loading acts as a dynamic force driving the disease by activating mechanoresponsive signaling, which triggers the production of pro-inflammatory mediators and catabolic enzymes through Mitogen-activated protein kinases (MAPK) pathway regulation [6,7]. Therefore, catabolic processes break down the functional extracellular matrix (ECM), leading to changes in the composition and viscoelastic properties of the ECM produced by chondrocytes. These alterations create an abnormal loading environment that promotes cell dysfunction and inflammation. The cleavage of the core proteoglycan molecule by aggrecanases leads to the loss of proteoglycans in OA, which is considered a key pathological feature of OA [8]. Articular cartilage degradation is also driven by matrix metalloproteinases (MMPs), notably MMP13, which plays a critical role in OA by breaking down type II collagen (COL2A1). In contrast, MMP3 primarily supports normal tissue turnover [9,10], while MMP1 and MMP9 synergistically degrade ECM components, amplifying tissue breakdown [11]. Moreover, in OA, chondrocytes can accumulate reactive oxygen species (ROS), resulting in cellular damage and inflammation [12]. Additionally, elevated levels of nicotinamide adenine dinucleotide (NADH) and lactate, along with alterations in glutamate and formate metabolism, disrupt cellular energy production and accelerate cartilage degeneration. These processes may also be influenced by mechanical loading [13].

Exercise therapy combined with weight loss is a widely recommended first-line treatment for OA, particularly effective in early stages for pain relief without the side effects of medication [14,15]. In daily life, tissues of the joint experience cyclically varying mechanical loads, alternating between periods of rest (e.g., sleep), moderate activity (e.g., walking), and higher stress (e.g., sports). Via a dynamic loading protocol varying in both intensity and duration, with low-strain phases representing rest periods and medium- to high-strain phases mimicking everyday movements and peak loads, one might mimic the situation of mechanical input on the cellular level. Furthermore, this approach better reflects physiologically relevant mechanical variability than constant stimulation and allows study of cellular adaptation under realistic conditions. The moderate SM/SA loading profile was designed to reflect natural movement patterns where phases of slow-moving (SM) activity alternate with phases of strong activity (SA) [16]. While many studies have explored mechanical stimulation in OA chondrocytes, inconsistent protocols and cell sources limit generalization [17,18,19,20]. Notably, juvenile and adult bovine chondrocytes show age-related differences in catabolic responses to loading [21]. Data on the effects of mechanical stimulation in human juvenile healthy chondrocytes remain lacking.

In this study, we aim to address the existing knowledge gap regarding the differences in expression profiles, metabolic components, and MAPK phosphorylation between juvenile and OA chondrocytes, specifically in response to physiological mechanical loading. We hypothesized that juvenile and OA-chondrocytes differ in baseline Biological Outcome Measurement (BIOM) expression and exhibit divergent transcriptional, metabolic, and kinase phosphorylation responses to cyclic tensile strain.

## 2. Results

### 2.1. Gene Expression Analysis of Relevant Biomarkers in Unstimulated Chondrocytes

As illustrated in the workflow (Figure 1a), jCH and pCH-OA were isolated from nine human cartilage donations each and used in passages 1 and 2 for the subsequent experiments. The initial step was to assess the differences in gene expression between juvenile, healthy chondrocytes and those exhibiting the OA phenotype using qPCR analysis. The measured variables consisted of 19 biomarkers (BIOMs), categorized as catabolic, hypertrophic, inflammatory, and anabolic markers (Figure 1b).

Each BIOM was assessed three times per donor to reduce measurement error. An a priori power analysis using GPower software (v3.1) indicated that *n* = 9 per group is sufficient to detect a large effect size (Cohen’s d ≥ 0.8) with α = 0.05 and power = 0.80. Gene expression analysis revealed that collagen I (COL1) and Runt-related transcription factor 2 (RUNX2) were expressed at higher levels in jCH cells compared to pCH-OA cells, as evidenced by lower (more negative) ΔCT values. In contrast, MMP3, SOX5, and aggregan (ACAN) exhibited reduced expression in jCH cells. Table 1 presents the mean values ± standard deviations (SD) along with statistical analyses using Levene’s test, Student’s *t*-test, and Cohen’s d. A lower p-value in the Levene’s test indicates more comparable or homogeneous variances between the two comparison groups, with SOX5 having the lowest *p*-value, followed by ACAN and RUNX2. The effect sizes measured with Cohen’s d quantify how strongly jCH and pCH-OA chondrocytes differ in BIOM expression under baseline conditions, with higher values reflecting greater differences. Values > 0.8 indicate a strong effect, observed across all listed BIOMs within Table 1. The graphical representation of the significant BIOMs using ΔCt values is shown in Figure 2.

The results of the PCA, shown in Figure 3, reveal the relationships among the 19 BIOMs, which were grouped into 5 factors through factor analysis. Each factor can be seen as a latent variable that influences several of the BIOMs, highlighting how they differ in the patterns of human jCH vs. pCH-OA. Nevertheless, both groups showed a good model fit of the five-factor solution found, with 92.8% of the variance explained for jCH and 91.3% for pCH-OA, meaning that the 19 BIOMS can be represented very efficiently by five latent dimensions. The individual BIOMs showed a high commonality [h2], with values > 0.90 common variance in both groups. Individual exceptions were osteopontin (SPP) in jCH with a lower h2 value of 0.75 (Figure 3a) and in pCH-OA a reduced h2 in MMP1 and SOX5 (Figure 3b). The descriptive-qualitative comparison of the two Principal Component Analysis (PCA) models demonstrated that the correlations between the gene expressions of the individual BIOMs differed significantly between healthy jCH and those with the OA phenotype. CCAAT/enhancer binding proteins (cEBP), interleukin 8 (IL8), ADMTS4 and COL1 showed high loadings on a latent dimension in jCH (see factor 2), whereas SOX6, ADMTS5, SOX5, and IL6 revealed dependencies in their gene expression in pCH-OA. MMP3, which shows a significant difference in expression between the two cell types, appears to be associated with the bone morphogenetic protein 2 (BMP2), and both cluster in pCH-OA in factor 1, an effect not seen in jCH. Similarly, pro-inflammatory BIOMs (IL6, IL8) tend to cluster together, with a higher grade of relevance in pCH-OA, whereas osteocalcin (BGLAP) and MMP1 are not differentially distributed between the two cell types at all. The intercorrelations between BIOM expression in jCH often differ markedly from those in pCH-OA (Appendix A). BGLAP shows a negative correlation (inverse relationship) with MMP1 and IL6 in jCH cells, while it was identified positively correlated in pCH-OA cells. Similarly, MMP3 versus MMP1 and cEBP versus COL1 exhibit different correlation patterns between jCH and pCH-OA cells (Appendix A).

### 2.2. Effect of Mechanical Stimulation on the Relative Gene Expression of Relevant Chondrocytes Biomarkers

To investigate how mechanical forces influence chondrocyte behavior, jCH and pCH-OA were exposed to cyclic tensile strain, allowing us to assess the effects of dynamic mechanical stimulation on cellular phenotype, matrix production, and mechanoresponsive signaling pathways. After 48 h of mechanical stimulation using the SM/SA profile, jCH and pCH-OA were harvested, total RNA was isolated and the relative gene expression of 19 BIOMs was analysed by qPCR. Gene expression of MMP1, ADAMTS-4/5, ACAN, BMP2, and SPP1 was significantly increased in both cell types when comparing mechanically stimulated cells to unstimulated controls. In contrast, MMP3, RUNX2, SOX5/9 and the Toll-like-Receptor 4 (TLR4) were upregulated exclusively in jCH. With regard to IL6 and IL8, only the pCH-OA showed a response to mechanical stimulation in the form of a clear downregulation. No significant differences were detected in the expression levels of MMP13, COL1A1, BGLAP, SOX6, and cEBP. All data can be found in the Appendix A.

Considering the baseline differences in gene expression between the two cell types, we further examined the variations in magnitude and response to mechanical stimulation in jCH and pCH-OA. The expression of BMP2 and MMP1 was significantly upregulated in response to mechanical stimulation, with the extent of this increase differing between the two cell types. Conversely, the expression of RUNX2, IL6, IL8, and TLR4 was regulated in opposite directions by mechanical stimulation in the two cell types. All other BIOMs responded similarly, showing no significant differences between the cell types. The corresponding mean values ± SD, and the statistical evaluations using Levene test, student’s *t*-test, and Cohen’s d are shown in Table 2. The effect size (Cohen’s d) represents the extent of differences in BIOM expression between jCH and pCH-OA under varying mechanical loading conditions, with higher values indicating greater differences. A corresponding graphical representation is presented in Figure 4.

### 2.3. Impact of Cyclic Tensile Strain on Chondrocyte Metabolism

To further understand the cellular response to mechanical stimulation, metabolic parameters of jCH and pCH-OA were analyzed, providing insights into energy utilization and matrix synthesis under dynamic loading conditions. jCH and pCH-OA were seeded on BioFlex^®^ plates and stimulated for 48 h using the SM/SA profile to assess the effects of cyclic tensile strain on metabolism. Metabolic assays were performed on collected cultures and lysates. The measurement of hydrogen peroxide (H_2_O_2_) demonstrated a significant reduction in response to the mechanical stimulation, with a similar effect observed in both cell types. The luminescence values (mean ± SEM) were reduced from 5480 ± 880 to 3798 ± 1045 (**) in the case of jCH and from 5903 ± 910 to 4499 ± 862 (**) for pCH-OA. The Menadione positive control was 15 times higher, confirming the effectiveness of the assay (Figure 5a). In both chondrocyte types, the application of cyclic tensile strain resulted in a highly significant reduction in NAD^+^/NADH levels more pronounced in pCH-OA and with a significant difference observed between the two cell lines (Figure 5b). The analysis of the three key metabolites in cellular metabolism glutamate (Figure 5c), lactate (Figure 5d), and formate (Figure 5e) revealed a significant increase in response to mechanical stimulation compared to the unstimulated control cells. No differences in metabolic activity were observed between jCH and pCH-OA.

### 2.4. Cyclic Tensile Strain Alters Protein Phosphorylation

Phosphorylation levels of key signaling pathways were analyzed to assess how cyclic mechanical loading modulates mechanoresponsive signaling in jCH and pCH-OA. Protein phosphorylation levels of key regulators were assessed via immunoblotting under unstimulated control conditions (control) and after 48 h stimulation with the SM/SA profile in both jCH (Figure 6a) and pCH-OA (Figure 6b). Under these experimental conditions, cyclic tensile strain led to a significant decrease in the phosphorylation of the serine/threonine kinase AKT as well as in the ERK/JNK/p38 signaling pathway in juvenile chondrocytes, while STAT3 phosphorylation showed a significant increase. The OA phenotype aligned with the healthy juvenile chondrocytes in AKT, while STAT3 and JNK/p38 showed opposing patterns, with a notable increase in JNK/p38 phosphorylation observed while ERK did not seem affected.

## 3. Discussion

OA is the most common degenerative joint disease, with no current cure or remission-inducing treatments. Research focuses on identifying biomarkers to better understand disease mechanisms. Signaling proteins are linked to cartilage degradation, anabolism, inflammation, and hypertrophy. Catabolic enzymes like MMPs and aggrecanases are elevated in OA [22], while anabolic markers (COL, ACAN, BMPs) and SOX5/6/9 promote cartilage maintenance [23,24,25]. Hypertrophic markers (RUNX2, BGLAP, SPP1, MMP13) indicate chondrocyte maturation [26], and TLR4 activation drives inflammatory cytokines such as IL6 and IL8 [27]. This study aimed to compare gene expression profiles between jCH and pCH-OA, and to assess their distinct responses to physiological mechanical stimulation. The goal was to identify expression differences that may reflect altered regulatory pathways in pCH-OA. Cartilage tissue from children served as a healthy reference for OA patients, since the risk of OA in this population is minimal, allowing the tissue to be regarded as healthy. A potential limitation of this study is that metabolic differences between age groups may influence the results, complicating direct comparisons between juvenile and OA chondrocytes. It should also be taken into consideration that age represents one of the main risk factors for OA. Therefore, a comparison between young and old chondrocytes is meaningful and highly relevant, as specific genes may be differentially expressed in an age-dependent manner. Chondrocytes from juvenile donors show elevated expression of chondrogenic genes, like COL2A1 or ACAN, resulting in stronger ECM production compared to adult and OA chondrocytes [28].

Our data revealed significantly higher MMP3, SOX5, and ACAN, but lower COL1 and RUNX2 levels in pCH-OA compared to jCH. PCA highlighted clear differences in gene expression patterns and interactions among the 19 BIOMs, indicating distinct molecular profiles for each cell type. Factor analysis further showed differing groupings, with a notable stronger association between MMP3 and BMP2 in pCH-OA. In jCH, BMP2 is essential for proliferation, matrix production, and differentiation, particularly during endochondral ossification and cartilage repair. It regulates key genes like SOX9 and supports chondrocyte growth and maturation [29]. In OA, BMP2 levels in serum and synovial fluid correlate with radiographic and symptomatic severity, indicating its potential as a biochemical marker for disease severity [30]. Additionally, SOX9 may enhance BMP2-driven chondrogenesis by suppressing Smad7, offering promise for cartilage tissue engineering [31]. Our factor analysis comparing jCH and pCH-OA reveals a disrupted relationship between BMP2 and SOX9, with BMP2 shifting to factor 1 and SOX9 to factor 5. Notably, BMP2 now shows a stronger association with MMP3, suggesting a new interaction in OA pathology. Additionally, the shift of IL6 and IL8 to higher priority indicates heightened inflammatory processes in pCH-OA, consistent with inflammatory OA phenotypes. Our analysis concludes that as chondrocytes transition from a normal to an OA phenotype, changes in signaling pathways occur, altering the relationships between key molecules and affecting the stability of these pathways. Drawing conclusive statements about chondrocyte responses to mechanical stimuli from the literature is difficult, as studies vary considerably in loading intensity, duration, and frequency [32,33,34]. To address this issue, we used human primary cells and a well-established physiological stimulation protocol that mimics natural movement and includes periods of increased loading, to achieve clinically relevant results. This approach enables the direct comparison of jCH and pCH-OA and utilizes the unique value of jCH as a rare, non-OA reference. Significant attention has been focused on understanding how dysfunction induced by mechanical stress contributes to the pathogenesis of OA. Articular cartilage experiences dynamic mechanical loads, which chondrocytes detect and respond to. While physiological loading enhances chondrocyte metabolism and matrix synthesis [35,36], abnormal loading disrupts specific signaling pathways, accelerating cartilage degeneration and promoting OA development [37,38].

The second part of our study examined the effect of physiological cyclic tensile strain with short periods of increased exertion, as occurs in exercise therapy, on human chondrocyte metabolism. Our definition of physiological stimulation refers to mechanical stimuli that reflect healthy joint movement and promote cartilage maintenance. Using a cyclic tensile strain with alternating low and high loads and resting phases, we observed significant differences in the expression of BMP2, IL8, MMP1, RUNX2, and TLR4 between juvenile and OA chondrocytes. Factor analysis links BMP2 to MMP3 and IL8 to RUNX2, whose expression declines during inflammation [39]. The reduction in inflammatory markers is particularly important, as inflammation alters chondrocyte metabolism, and this metabolic reprogramming may be a key factor in OA progression.

Next to gene expression changes free radical production is elevated in OA patients, with oxidative stress playing a key role in articular cartilage degradation and being closely linked to aging and OA [40,41]. In chondrocytes, ROS acts as a signal transducer for differentiation and ECM synthesis and is regulated in response to mechanical stimulation. Excessive ROS production caused by inflammation leads, among other things, to mitochondrial dysfunction and cartilage degeneration. ROS amplifies inflammation in OA by activating pathways like MAPK and PI3K/AKT [42,43], raising pro-inflammatory cytokines and MMPs that damage cartilage. ROS also increases tissue inhibitors of MMPs (TIMPs), creating an imbalance that accelerates extracellular matrix breakdown. However, physiological mechanical loading, such as the SM/SA profile, reduces ROS generation in both jCH and pCH-OA, suggests that exercise therapy may positively influence the progression of OA. Chondrocyte metabolism depends on glycolysis, oxidative phosphorylation, and other metabolic pathways to generate energy. These cells can utilize various substrates, including simple sugars, amino acids, and fatty acids found in the synovial fluid of joints [44]. However, chondrocyte metabolism is not solely dedicated to energy production; it also plays a crucial role in synthesizing ECM components through anabolic processes. A decreased NAD+/NADH concentration often signifies disrupted cellular respiration, elevated oxidative stress, or metabolic dysfunctions. This leads to a shift toward anaerobic metabolism and heightened lactate production (Warburg effect) [45]. Both can be observed after mechanical stimulation in both jCH and pCH-OA. Cyclic tensile strain significantly increased the levels of key metabolic intermediates in jCH and pCH-OA. Glutamate, lactate, and formate are markedly elevated in response to mechanical loading. This metabolic shift reflects enhanced glycolytic activity, redox adaptation, and increased anaplerotic flux through amino acid and one-carbon metabolism, suggesting that mechanical stimulation substantially alters chondrocyte energy metabolism and cellular signaling. Additionally, ROS can act as secondary messengers, resulting in the increased activation of redox-sensitive pathways, including the mitogen-activated protein kinases (MAPKs) such as ERK1/2, the p38 cascade, and c-JNK. Phosphorylation serves as a molecular switch, either activating or inhibiting protein activity, depending on the context and the specific protein involved. This dynamic modification allows cells to respond rapidly to external signals, such as growth factors, hormones, or mechanical stimuli, by coordinating complex signaling networks. MAPK pathways have been found to be activated in OA cartilage by extracellular stimuli like pro-inflammatory cytokines, growth factors, and mechanical stress, and there is evidence suggesting that, at least for ERK, these pathways may play a crucial role in the cartilage degradation observed in OA [46]. Therefore, they represent key factors in OA pathogenesis [47,48]. In our cell systems, mechanical stimulation with the SM/SA profile led to a significant decrease in p-AKT and p-ERK, while p-p38 showed a trend toward an increase. The opposite regulation pattern of p-STAT and p-JNK between jCH and pCH-OA likely reflects differences in cellular context, where chronic inflammation, altered redox balance, and age- or disease-related epigenetic modifications in OA cells reprogram signaling pathways, leading to cell type specific divergent responses to the same stimulus. As previously discussed, the altered interactions and affinities among variables (BIOMs) that constitute different factors in jCH versus pCH-OA may also impact critical components of signaling pathways, particularly kinases. For instance, BMP2, through its engagement with the STAT and JNK pathways, may contribute to reshaping kinase regulation by modifying its associations with other relevant proteins. Reduced AKT and ERK phosphorylation in cartilage cells could lead to degeneration and reduced regenerative capacity. STAT3 can be activated by several cytokines, with IL6 being the most representative. IL6 induces a reduction in nuclear export signals, which then regulates the transcriptional processes of downstream target genes through the modulation of growth factors [49]. However, further experiments using pharmacologic inhibitors or gene silencing approaches are needed to elucidate the functional role of MAPK phosphorylation changes.

## 4. Materials and Methods

### 4.1. Cartilage Samples and Cell Culture of Primary Cells

This study was conducted at the Department of Orthopedics and Traumatology, Medical University of Graz, in accordance with ethical standards (ethics approvals: 35-489ex22/23 for primary OA chondrocytes (pCH-OA), valid until 13 October 2025; 35-278ex22/23 for juvenile chondrocytes (jCH), valid until 2 May 2025; both annually renewable) and the Declaration of Helsinki. pCH-OA were isolated from femoral cartilage of nine patients (mean age 68.2 ± 7.3 years) undergoing end-stage knee arthroplasty, with informed consent. Tissue was digested in growth medium with 2 mg/mL collagenase B at 37 °C for 24 h, followed by filtration (40 µm) and centrifugation; passages 1–2 were used for experiments. jCH were obtained from nine patients (mean age 14.7 ± 4.5 years) undergoing knee arthroscopy after patellar dislocation with flake fracture, with informed consent from patients or guardians. The same isolation protocol was used. To minimize dedifferentiation, cells were cultured during the expansion period in DMEM/F12 medium supplemented with 10% FBS, 1% Penicillin-Streptomycin (5000 U/mL), 1% L-Glutamine, 1% Insulin-Transferrin-Selenium, 0.01% TGF-β (1 ng/mL), and 0.01% FGF (1 ng/mL). Cultures were maintained at 37 °C in a humidified 5% CO_2_ atmosphere.

### 4.2. Mechanical Stimulation of Chondrocytes

Mechanical stimulation of chondrocytes was performed using the Flexcell™ FX5K Tension System (Flexcell International, Burlington, NC, USA), applying vacuum-driven strain to cells cultured on pronectin-coated BioFlex^®^ membranes in a six-well format. BioFlex^®^ plates feature a flexible silicone membrane that enables the application of controlled tensile strain to the cultured cells. Chondrocytes (5 × 10^4^ cells/well) were subjected to the SM/SA strain protocol, consisting of an 8 h rest followed by four cycles of alternating 2 h of slow cyclic strain (SM, slow-moving; 0.2 Hz, 2% elongation) and 2 h of high-intensity strain (SA, strong activity; 0.5 Hz, 15% elongation) over 48 h (Figure 1a). Control cells were cultured under identical conditions without mechanical stimulation to establish baseline responses. This protocol enables a controlled comparison of mechanotransductive responses between jCH and pCH-OA, assessing how cyclic tension influences transcriptional programs, metabolic activity, and kinase signaling. Alternating low- and high-intensity strains allows identification of strain-dependent and cell type-specific molecular responses, shedding light on the mechanisms underlying cartilage adaptation and degeneration.

### 4.3. Reverse Transcription Polymerase Chain Reaction

Following mechanical stimulation, total RNA was extracted using the RNeasy Mini Kit with DNase-I treatment (Qiagen, Hilden, Germany). Two micrograms of RNA were reverse-transcribed using the iScript cDNA Synthesis Kit with a mix of oligo(dT) and random hexamer primers. qPCR was performed in technical triplicates using SsoAdvanced Universal SYBR Green Supermix on a CFX96 Touch Real-Time PCR System (BioRad Laboratories Inc., Hercules, CA, USA), with QuantiTect primers (Qiagen) for target genes. CT values were analysed using CFX Manager (v3.1), excluding values > 32. Relative gene expression was calculated via the ∆∆Ct method, normalized to the geometric mean of β-2-microglobulin (B2M) and TATA-box binding protein (TBP). Gene expression was normalized to reference genes (ΔCT), and changes relative to untreated controls were calculated using the 2^−ΔΔCt^ method. Baseline comparisons were also derived from ΔCT values. QuantiTect primer assays (Qiagen) were used to analyze the expression of the matrix metalloproteases (MMPs)1, 3, 13, ADAMTS-4, ADAMTS-5, type I collagen (COL1A1), collagen X (ColX), aggrecan (ACAN), bone morphogenetic protein 2 (BMP2), Runt-related transcription factor 2 (RUNX2), osteopontin (SPP1), osteocalcin (BGLAP), SOX5, SOX6, SOX9, interleukins IL6, IL8, and the Toll-like-Receptor 4 (TLR4), and CCAAT/enhancer binding proteins (cEBP).

### 4.4. Metabolic Assays

Reactive oxygen species (ROS) were quantified using the ROS-Glo™ H_2_O_2_ assay (Promega, Madison, MI, USA) on supernatants collected after 48 h of SM/SA mechanical stimulation, with luminescence values normalized to controls and H_2_O_2_ levels determined via a standard curve. NAD^+^/NADH ratios were measured using the NAD/NADH-Glo™ assay, and glutamate levels were assessed with the Glutamate-Glo™ assay (1:40 dilution), both following the manufacturer’s protocols. Lactate was quantified using the Lactate-Glo™ assay (all Promega), and formate levels were measured with the Formate Assay Kit (Abcam, Cambridge, UK), based on a colorimetric signal at 450 nm. All assays were performed in triplicate, and results are reported as mean ± SD.

### 4.5. Protein Expression and Phosphorylation Analysis

Whole cell protein extracts were prepared immediately after mechanical stimulation with the SM/SA profile using lysis buffer (50 mM Tris-HCl pH 7.4, 150 mM NaCl, 1 mM NaF, 1 mM EDTA, 1% NP-40, 1 mM Na_3_VO_4_) and a protease inhibitor cocktail (Sigma Aldrich, St. Louis, MI, USA). Protein concentration was determined using the Pierce BCA Protein Assay Kit (Thermo Fisher Scientific, Waltham, MA. USA). The proteins were separated by SDS-PAGE and transferred to Amersham™ Protran™ Premium 0.45 µM nitrocellulose membranes (GE Healthcare Life Science, Little Chalfont, UK). Primary antibodies against the p-AKT/AKT, p-STAT3/STAT3, p-ERK/ERK, p-JNK/JNK, and p-p38/p38 were incubated overnight, and β-actin (all from Cell Signaling Technology, Danvers, MA, USA) was used as loading control. Blots were incubated with a horseradish peroxidase-conjugated secondary antibody (Dako) for 1 h, followed by detection with the Amersham™ ECL™ Prime Western blotting detection reagent (GE Healthcare). Chemiluminescence signals were captured using the ChemiDocTouch Imaging System and images were analysed with ImageLab 5.2 Software (Bio-Rad Laboratories Inc.).

### 4.6. Statistical Analysis

The mean values and standard deviations were calculated for the relative gene expression of the ΔCt values (Biological Outcome Measurements, or BIOMs). Student’s *t*-tests were performed to compare the two groups (jCH vs. pCH-OA) and evaluate the effect of physiological mechanical stress. These calculations used the mean value of three determinations per donor. Missing values (*n* = 1) were excluded from the statistical analysis. Assumptions such as normal distribution and homogeneity of variance were checked. The Levene test was used to verify equality of variance, and the degrees of freedom were corrected as necessary (Welch *t*-test). Effect sizes (Cohen’s d) of |d| = 0.2 are considered small, |d| = 0.5 are considered medium, and |d| ≥ 0.8 are considered large. *p*-values below 0.05 (two-sided) were considered statistically significant. With nine samples per group, the sample size is sufficient to demonstrate a large interaction effect at a 5% significance level (α = 0.05) and 80% statistical power (1 − β = 0.80). Due to the small sample size and power estimation, no correction was made for the multiple comparisons (i.e., multiple *t*-tests) between the different BIOMs. The evaluations should therefore be considered hypothesis-generating. The same exploratory approach also applies to the robustness of the principal component analysis (PCA), which was performed to assess the linear relationship between gene expression and the different BIOMs, and to obtain a smaller set of uncorrelated latent variables, known as principal components, with as little loss of information as possible. The results of two PCAs were descriptively compared to generate hypotheses and explore potentially similar or different functional relationships between the gene expression of jCH and pCH-OA. All calculations were performed using IBM SPSS Statistics 29.

## 5. Conclusions

In our study, we compared juvenile chondrocytes with OA chondrocytes, uncovering distinct differences that suggest altered signaling pathways in pCH-OA. These observations offer important insights into the molecular alterations associated with the onset and progression of OA, potentially shedding light on the mechanisms underlying OA pathogenesis. Additionally, we characterized juvenile chondrocytes at the expression level, making this one of the first studies to do so. Notably, we observed changes in the interaction between BMP2 and SOX9, as well as alterations in MMP3 expression and inflammatory cytokines such as IL6 and IL8 in pCH-OA. These insights may inform new therapeutic strategies, particularly in using exercise-based interventions to counteract OA progression.

## Figures and Tables

**Figure 1 ijms-26-10934-f001:**
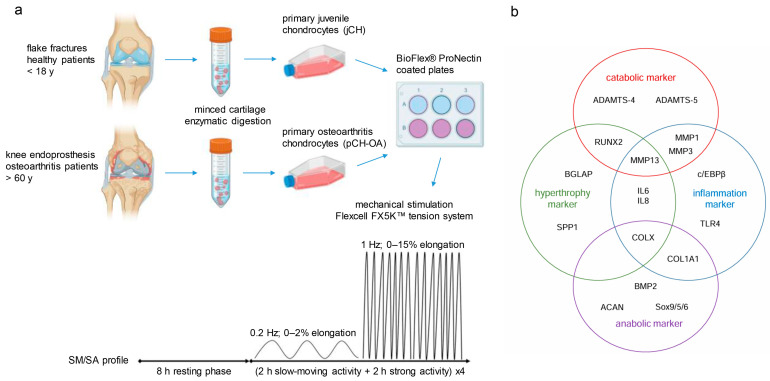
(**a**) Graphical representation of the project workflow about the origin of the primary cells and the process of mechanical stimulation. (**b**) Categorisation of the analysed biomarkers (Biological Outcome Measurements; BIOMs) in catabolic, hypertrophic, inflammatory, and anabolic markers.

**Figure 2 ijms-26-10934-f002:**
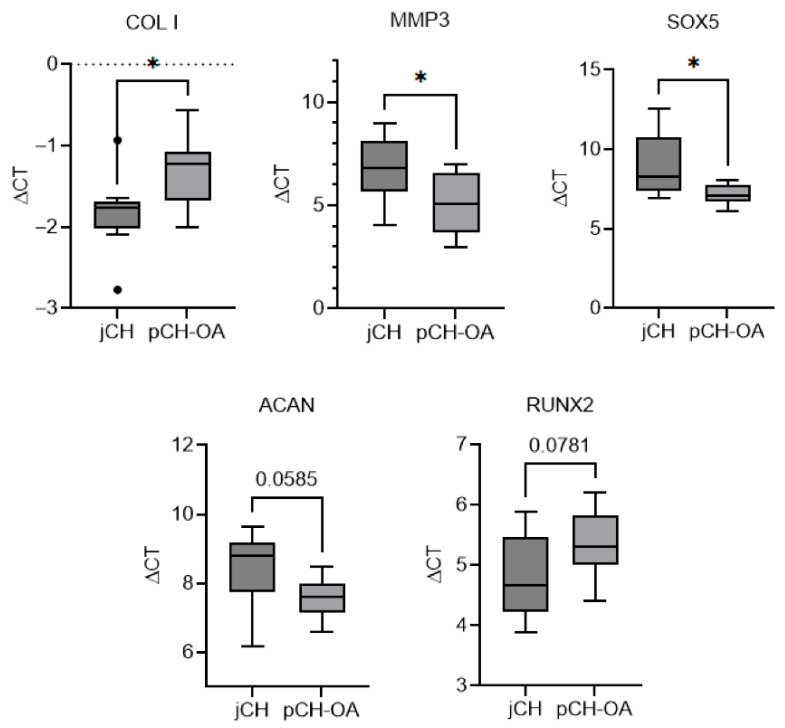
Graphical representation of significantly different gene expression patterns of ACAN, COL1, MMP3, RUNX2, and SOX5 genes between human jCH and pCH-OA under unstimulated conditions. Data are presented as box-and-whisker plots (*n* = 9; measured in biological triplicates). Groups were compared using the unpaired, parametric Student’s *t*-test. Statistical significance is indicated as follows: *: *p* < 0.05; trends toward significance for differences between two BIOMs are noted at *p* < 0.1.

**Figure 3 ijms-26-10934-f003:**
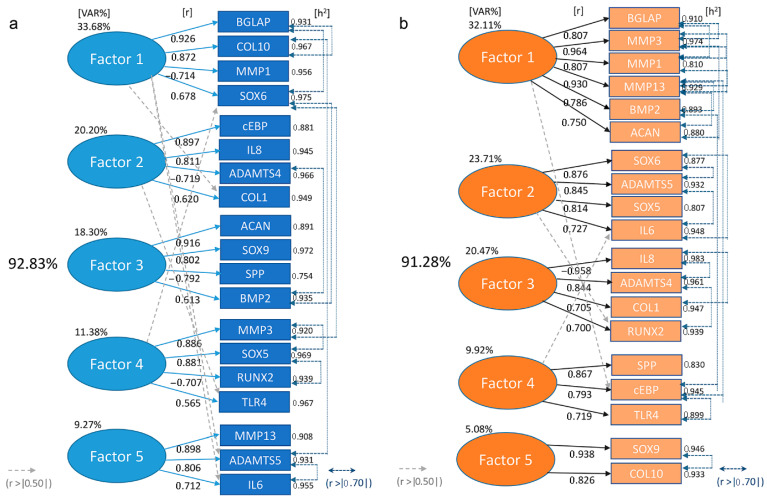
Principal Component Analysis (PCA) of (**a**) jCH and (**b**) pCH-OA chondrocytes under unstimulated conditions revealed fundamental differences in the expression and interaction of the 19 BIOMs between the cell types. The following abbreviations were used: explained variance [VAR%]; communality [h2]; arrows and dashed lines indicated higher correlation coefficients [r].

**Figure 4 ijms-26-10934-f004:**
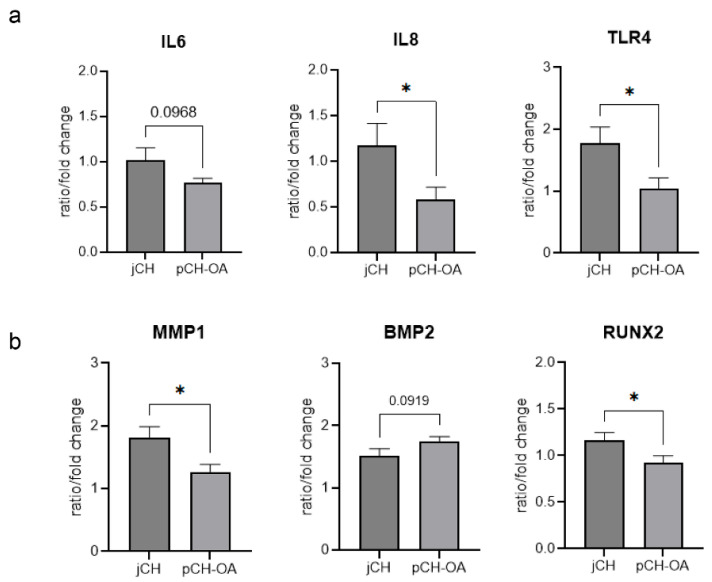
Graphical representation of significantly different gene expression patterns of (**a**) the inflammation markers IL6, IL8, and TLR4, and (**b**) MMP1, BMP2, and RUNX2 genes between jCH and pCH-OA under 48 h mechanical stimulation with the SM/SA profile. Mean ± SD (*n* = 9; measured in biological triplicates). Statistical significance is indicated as follows: *: *p* < 0.05; trends toward significance for differences between two BIOMs are noted at *p* < 0.1.

**Figure 5 ijms-26-10934-f005:**
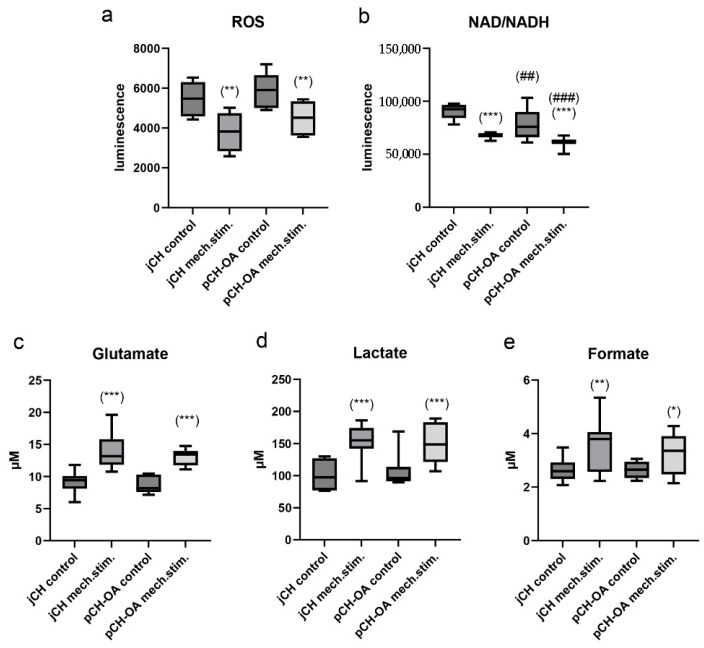
Impact of mechanical stimulation on chondrocyte metabolism. (**a**) ROS; (**b**) NAD^+^/NADH, (**c**) glutamate; (**d**) lactate, and (**e**) formate. All measurements were conducted in triplicates, and results were expressed as mean ± SD. Statistically significant differences between unstimulated control cells and mechanically stimulated cells were presented with *: *p* < 0.05; **: *p* < 0.01; ***: *p* < 0.001; differences between jCH and pCH-OA were presented with ##: *p* < 0.01; ###: *p* < 0.001.

**Figure 6 ijms-26-10934-f006:**
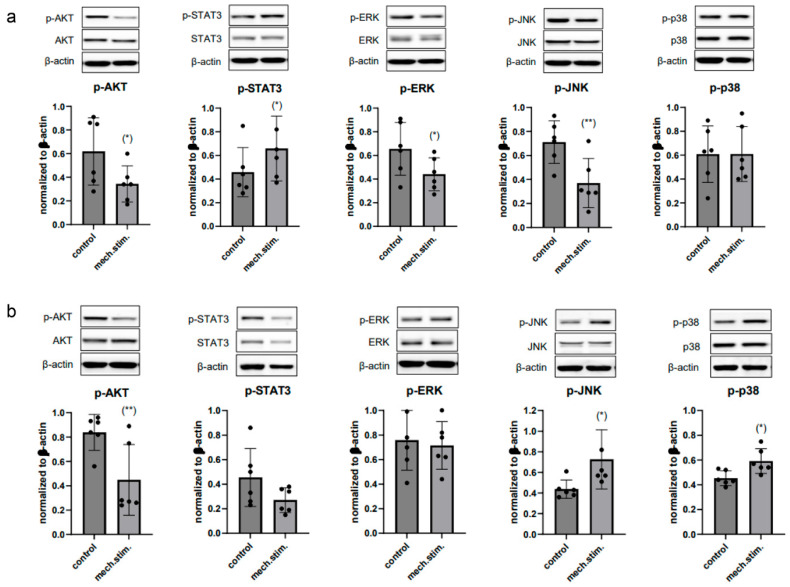
Cyclic tensile strain applied for 48 h using the SM/SA profile altered the protein phosphorylation of MAPKs in (**a**) jCH and (**b**) pCH-OA. The phosphorylation of proteins including AKT, STAT3, and ERK/JNK/p38 was assessed by immunoblotting under both unstimulated control conditions and after mechanical stimulation. β-actin served as the loading control (mean ± SD, *n* = 6). Statistical significance was defined as follows: * *p* < 0.05; ** *p* < 0.01.

**Table 1 ijms-26-10934-t001:** Significantly different gene expression patterns of jCH and pCH-OA.

BIOM	Group	n	Mean	SD	Levene Test [*p*]	*t*-Test [*p*]	Cohen’s d
*COL1*	jCH	9	−1.84	0.48	0.93	0.021 *	1.20
	pCH-OA	9	−1.30	0.43			
*MMP3*	jCH	9	6.73	1.55	0.75	0.039 *	1.06
	pCH-OA	9	5.10	1.51			
*SOX5*	jCH	9	9.03	1.98	0.00 **	0.024 *	1.26
	pCH-OA	9	7.18	0.64			
*ACAN*	jCH	9	8.44	1.11	0.15	0.058 (*)	0.96
	pCH-OA	9	7.59	0.57			
*RUNX2*	jCH	9	4.81	0.69	0.24	0.079 (*)	0.88
	pCH-OA	9	5.36	0.54			

Significant *t*-test with independent samples (groups: juvenile vs. OA) of gene expression analysis of 19 biomarkers (Biological Outcome Measurements; BIOMs) under baseline conditions. Statistically significant differences were presented with (*) *p* < 0.10, *: *p* < 0.05; **: *p* < 0.01.

**Table 2 ijms-26-10934-t002:** Significantly different gene expression patterns of jCH and pCH-OA under the influence of cyclic tensile strain with the SM/SA profile.

BIOM	Group	n	Mean Difference	SD	Levene Test [*p*]	*t*-Test [*p*]	Cohen’s d
*IL6*	jCH	9	0.91	0.25	0.077	0.097 (*)	0.83
	pCH-OA	9	0.76	0.16			
*IL8*	jCH	9	1.17	0.69	0.084	0.046 *	1.06
	pCH-OA	9	0.59	0.39			
*TLR4*	jCH	9	1.78	0.74	0.141	0.029 *	1.17
	pCH-OA	9	1.05	0.50			
*MMP1*	jCH	9	1.81	0.53	0.245	0.022 *	1.19
	pCH-OA	9	1.27	0.35			
*BMP2*	jCH	9	1.51	0.34	0.341	0.092 (*)	−0.85
	pCH-OA	9	1.75	0.21			
*RUNX2*	jCH	9	1.16	0.25	0.706	0.048 *	1.01
	pCH-OA	9	0.93	0.22			

Statistically significant differences were presented with (*) *p* < 0.10, *: *p* < 0.05.

## Data Availability

The original contributions presented in this study are included in the article/Appendix A. Further inquiries can be directed to the corresponding authors.

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
