# Peer review of "Juvenile and Osteoarthritic Human Chondrocytes Under Cyclic Tensile Strain: Transcriptional, Metabolic and Kinase Responses"

_ijms, 2025, doi:10.3390/ijms262210934_

Round 1
Reviewer 1 Report
Comments and Suggestions for Authors
The manuscript proposed by Lohberger et al. reports a comparison between primary juvenile and osteoarthritic human chondrocytes following in vitro simulation of mechanical stimulation in terms of genetic and metabolic signature.
The work is very interesting because in tissues physiologically highly subject to mechanical stress such as cartilage, the mechanobiology underlying cellular metabolism represents a new and interesting source for the study of degenerative diseases.
Overall, the work is well done with rigorous methodologies and well-described and discussed results. Consequently, only a few minor points need to be clarified before recommending the work for publication. Below are my comments:
- Although justified by the type of intervention, the age of young donors is rather low (14/15 years); this aspect should be taken into greater consideration in the discussion as age itself represents a key factor for the underlying conditions of the cellular signature.
- I fully agree with the authors' comment that it is very difficult to compare/comment on the parameters of the experimental set-up of mechanical stimulation, but it would be even clearer if the parameters chosen in this work were justified in more detail and not just with generic references to physiological exercise or physiotherapy treatment.
- I find the discussion on the pro-regenerative aspects of the pathways that differentiate the signatures of the two cell populations a bit neglected; this part could be expanded.
- I suggest a grammar check, there are still some typos such as lines 228-230 (from the guidelines, I guess).
Author Response
The manuscript proposed by Lohberger et al. reports a comparison between primary juvenile and osteoarthritic human chondrocytes following in vitro simulation of mechanical stimulation in terms of genetic and metabolic signature.
The work is very interesting because in tissues physiologically highly subject to mechanical stress such as cartilage, the mechanobiology underlying cellular metabolism represents a new and interesting source for the study of degenerative diseases.
Overall, the work is well done with rigorous methodologies and well-described and discussed results. Consequently, only a few minor points need to be clarified before recommending the work for publication.
Author´s response: Thank you very much for your time and effort in reviewing our manuscript. We are very pleased about the positive assessment. We have carefully addressed all your comments and hope that we have correctly understood and implemented your suggestions.
Below are my comments:
- Although justified by the type of intervention, the age of young donors is rather low (14/15 years); this aspect should be taken into greater consideration in the discussion as age itself represents a key factor for the underlying conditions of the cellular signature.
Author´s response: We are aware of this limitation. Unfortunately, obtaining sufficiently healthy cartilage from older OA patients—or from healthy adult donors—is not feasible. For this reason, we opted to use cartilage from children as a healthy comparison group, since the likelihood of osteoarthritis in this population is minimal, allowing the tissue to be considered ‘healthy’.
Line 263-272: We have added the following paragraph (and an additional reference) to the discussion where this issue is addressed: “Cartilage tissue from children served as a healthy reference for OA patients, since the risk of OA in this population is minimal, allowing the tissue to be regarded as healthy. A potential limitation of this study is that metabolic differences between age groups may influence the results, complicating direct comparisons between juvenile and OA chondrocytes. Chondrocytes from juvenile donors show elevated expression of chondrogenic genes, like COL2A1 or ACAN, resulting in stronger ECM production compared to adult and OA chondrocytes [28].
- I fully agree with the authors' comment that it is very difficult to compare/comment on the parameters of the experimental set-up of mechanical stimulation, but it would be even clearer if the parameters chosen in this work were justified in more detail and not just with generic references to physiological exercise or physiotherapy treatment.
Author´s response: Thank you for highlighting this. We have included the following paragraph in the introduction to clarify our choice of stimulation profile:
Line 69-76: In daily life, tissues experience cyclically varying mechanical loads, alternating between periods of rest (e.g. sleep), moderate activity (e.g. walking), and higher stress (e.g. sports). To simulate this, a dynamic loading protocol can vary in both intensity and duration, with low-strain phases representing rest periods and medium- to high-strain phases mimicking everyday movements and peak loads. This approach better reflects physiologically relevant mechanical variability than constant stimulation and allows study of cellular adaptation under realistic conditions. The moderate SM/SA loading profile was designed to reflect natural movement patterns where phases of slow-moving (SM) activity alternate with phases of strong activity (SA) [16].“
- I find the discussion on the pro-regenerative aspects of the pathways that differentiate the signatures of the two cell populations a bit neglected; this part could be expanded.
Author´s response: We appreciate the reviewer’s observation that this section of the discussion required further elaboration. We have accordingly expanded this part to provide a more comprehensive interpretation of the results (Line 352-356).
- I suggest a grammar check, there are still some typos such as lines 228-230 (from the guidelines, I guess).
Author´s response: Thanks for pointing that out! Unfortunately, we overlooked that. It has been corrected.

Reviewer 2 Report
Comments and Suggestions for Authors
In this manuscript the authors investigate the effects of mechanical loading on healthy juvenile versus aged osteoarthritic chondrocytes. While the overall premise in studying healthy vs pathological chondrocytes in response to load is interesting and valuable, there are several significant concerns with the experimental design and interpretation of the data that limit the impact of the manuscript.
My biggest concern is the use of passaged chondrocytes. It is well established that even after 1 passage, chondrocytes are dedifferentiated and fibroblastic, losing their defining characteristics. Therefore, the study is not truly examining load responses in chondrocytes but rather in fibroblast-like cells. The lack of collagen type II (Col2) assessment further weakens the validation of these cells as representative of articular chondrocytes.
In addition, I am also concerned about their loading regimen. The authors apply 15% strain, which they consider physiological. However, in the cartilage mechanobiology field, 15% strain is generally considered pathological or overload. Physiological loading is typically modeled with <10% strain. This significantly impacts the interpretation of the results.
Below is a point-by-point critique:
- The comparison between “young” (juvenile) and “old + diseased” (osteoarthritic) tissue is unusual and confounds age with disease state. This makes it difficult to attribute observed effects specifically to age or pathology.
- For juvenile tissue, the anatomical source of the chondrocytes is not specified. Were cells harvested from the same joint type as the osteoarthritic samples? This information is critical for meaningful comparison.
- The exercise/loading protocol is problematic. A 15% strain is widely regarded as pathological rather than physiological, and most studies use <10% to model physiological loading.
- Use of passaged cells introduces dedifferentiation. Without Col2 or other chondrocytic markers, the identity of the cells is uncertain. The findings therefore may not represent true chondrocyte behavior.
- Not all abbreviations are written out in full upon first use.
- The manuscript discusses gene expression findings that are not shown in the figures. All reported results should be presented.
- The rationale for selecting specific metabolic factors is not adequately justified. Greater explanation of why these markers were chosen is needed.
- The results section does not have adequate transition statements making it difficult to follow the authors’ line of thought. Improved organization and flow would help.
- Lines 41–42: the sentence is unclear and should be rewritten for clarity.
- Lines 226–229: this section should be deleted and I am surprised they did not catch this during edits
- Line 260: the authors state there is a common difficulty in interpreting responses due to differences in loading protocol. This seems contradictory, as the loading protocol is a central part of their study and should have been carefully justified.
- Figure 4: was an unloaded control included? If not, one should be added, as this is necessary for interpreting the effects of load.
In summary, there are many serious methodological issues and major revisions are required before the manuscript can be considered for publication.
Author Response
In this manuscript the authors investigate the effects of mechanical loading on healthy juvenile versus aged osteoarthritic chondrocytes. While the overall premise in studying healthy vs pathological chondrocytes in response to load is interesting and valuable, there are several significant concerns with the experimental design and interpretation of the data that limit the impact of the manuscript.
My biggest concern is the use of passaged chondrocytes. It is well established that even after 1 passage, chondrocytes are dedifferentiated and fibroblastic, losing their defining characteristics. Therefore, the study is not truly examining load responses in chondrocytes but rather in fibroblast-like cells. The lack of collagen type II (Col2) assessment further weakens the validation of these cells as representative of articular chondrocytes.
In addition, I am also concerned about their loading regimen. The authors apply 15% strain, which they consider physiological. However, in the cartilage mechanobiology field, 15% strain is generally considered pathological or overload. Physiological loading is typically modeled with <10% strain. This significantly impacts the interpretation of the results.
Author´s response: Thank you very much for your time and effort in reviewing our manuscript. We have carefully addressed all your comments and hope that we have correctly understood and implemented your suggestions.
Below is a point-by-point critique:
- The comparison between “young” (juvenile) and “old + diseased” (osteoarthritic) tissue is unusual and confounds age with disease state. This makes it difficult to attribute observed effects specifically to age or pathology.
Author´s response: We are aware of this limitation. Unfortunately, obtaining sufficiently healthy cartilage from older OA patients—or from healthy adult donors—is not feasible. For this reason, we opted to use cartilage from children as a healthy comparison group, since the likelihood of osteoarthritis in this population is minimal, allowing the tissue to be considered ‘healthy’. Furthermore, we would like to point out that, when comparing these two entities, it is crucial to consider that age can modulate cellular metabolic processes and represents the strongest risk factor for the development of osteoarthritis. Consequently, this comparison inherently reflects not only the difference between healthy and diseased states, but also the influence of age, by contrasting young and old conditions.
- For juvenile tissue, the anatomical source of the chondrocytes is not specified. Were cells harvested from the same joint type as the osteoarthritic samples? This information is critical for meaningful comparison.
Author´s response: As described in the Methods section 4.1, the tissue samples for isolating jCH and pCH-OA cells were taken from the knee. “jCH were obtained from nine patients (mean age 14.7 ± 4.5 years) undergoing knee arthroscopy after patellar dislocation with flake fracture,…”
- The exercise/loading protocol is problematic. A 15% strain is widely regarded as pathological rather than physiological, and most studies use <10% to model physiological loading.
Author´s response: We fully agree that a permanent mechanical stimulation involving a 15% elongation cannot be considered physiological. In everyday life, tissues are exposed not to constant but cyclically varying mechanical loads, resulting from the natural alternation between rest phases (e.g., sleep), moderate activity (e.g., walking, sitting, light movement), and higher-intensity exertion (e.g., sports, physical labour). To experimentally simulate these conditions, a dynamic loading protocol can be applied, in which the mechanical stimulation is varied in both intensity and duration. Low-strain phases mimic periods of rest, while medium- and high-strain phases reproduce everyday movements and peak loads. Such a model more accurately reflects the physiologically relevant variability of mechanical stress than constant or uniform stimulation. This approach allows us for a better understanding of cellular adaptation responses under conditions that closely resemble real-life daily activity. In one of our publications from 2019 (Lohberger et al. Mechanical exposure and diacerein treatment modulates integrin-FAK-MAPKs mechanotransduction in human osteoarthritis chondrocytes. Cell Signal. 2019; 56:23-30. doi: 10.1016/j.cellsig.2018.12.010.) we tested various stimulation profiles and were able to show that our SM/SA profile led to a significantly decreased expression of the inflammatory marker IL-6 and the degenerative enzymes ADAMTS3, MMP1, MMP3, and MMP13.
For this reason, we decided to use the SM/SA profile. We have added a corresponding paragraph to the introduction (line 69-76).
- Use of passaged cells introduces dedifferentiation. Without Col2 or other chondrocytic markers, the identity of the cells is uncertain. The findings therefore may not represent true chondrocyte behavior.
Author´s response: To circumvent this critical aspects we were using very early passages (1–2) to ensure sufficient cell numbers. In addition, the culture medium was supplemented with TGF-β and FGF-2, both of which are known to minimize chondrocyte dedifferentiation and to help maintain the characteristic chondrocyte phenotype. TGF-β has been reported to promote the synthesis of cartilage-specific extracellular matrix components, such as collagen type II and aggrecan, while inhibiting the transition to a fibroblastic phenotype. FGF-2 on the other hand supports chondrocyte proliferation without inducing dedifferentiation, and when combined with TGF-β, it helps preserve the cells’ differentiated state.
- Not all abbreviations are written out in full upon first use.
Author´s response: Thank you for highlighting this. That has been corrected.
- The manuscript discusses gene expression findings that are not shown in the figures. All reported results should be presented.
Author´s response: Table S2 presents data on the effect of mechanical stimulation on jCH and pCH-OA. The effect between unstimulated control cells and mechanically stimulated cells is represented as fold changes (gained via the DDCT method) in mean values and standard deviations are given. Statistical significance of the effect is assessed using a Student's t-test and is indicated in the final column. We also investigated the differences in strength and response to mechanical stimulation between the two cell types, which are shown in Table 2. The expression of BMP2 and MMP1 was significantly upregulated in response to mechanical stimulation, with the extent of this increase differing between the two cell types. Conversely, the expression of RUNX2, IL6, IL8 and TLR4 was regulated in opposite directions by mechanical stimulation in the two cell types. All other BIOMs responded similarly and showed no significant differences between cell types. Further details were provided in the manuscript to clarify these confusions.
- The rationale for selecting specific metabolic factors is not adequately justified. Greater explanation of why these markers were chosen is needed.
Author´s response: We appreciate the reviewer’s observation that this section of the discussion required further elaboration. We have accordingly expanded this part to provide a more comprehensive interpretation of the results.
- The results section does not have adequate transition statements making it difficult to follow the authors’ line of thought. Improved organization and flow would help.
Author´s response: Transitional sentences have been added to enhance the overall flow and coherence of the text.
- Lines 41–42: the sentence is unclear and should be rewritten for clarity.
Author´s response: We have rephrased the sentence as follows: “Disturbance of this balance, particularly under excessive stress, contributes directly to the development and progression of osteoarthritis (OA).”
- Lines 226–229: this section should be deleted and I am surprised they did not catch this during edits
Author´s response: Thanks for pointing that out! We embarrassingly overlooked that.
- Line 260: the authors state there is a common difficulty in interpreting responses due to differences in loading protocol. This seems contradictory, as the loading protocol is a central part of their study and should have been carefully justified.
Author´s response: With this statement, we aimed to highlight that the term “mechanical stimulation” in the literature often encompasses a wide range of experimental parameters. At the same time, it is evident from numerous studies that substantial differences exist in loading magnitude, duration, and frequency, which are frequently difficult to compare.
We have revised the wording and hope that the passage is now clearer.
- Figure 4: was an unloaded control included? If not, one should be added, as this is necessary for interpreting the effects of load.
Author´s response: Yes, an unloaded control was included, as it would otherwise not be possible to evaluate the fold change (i.e., the ratio) induced by mechanical stimulation. The unloaded control therefore served as the reference group to which the “treated” samples were normalized using the ΔΔCT method.
